# Nanocrystal facet modulation to enhance transferrin binding and cellular delivery

Yu Qi[1,2,3], Tong Zhang [1✉], Chuanyong Jing [2,3], Sijin Liu[2,3], Chengdong Zhang[1,4], Pedro J.J. Alvarez [5✉] & Wei Chen[1]

Binding of biomolecules to crystal surfaces is critical for effective biological applications of crystalline nanomaterials. Here, we present the modulation of exposed crystal facets as a feasible approach to enhance specific nanocrystal–biomolecule associations for improving cellular targeting and nanomaterial uptake. We demonstrate that facet-engineering significantly enhances transferrin binding to cadmium chalcogenide nanocrystals and their subsequent delivery into cancer cells, mediated by transferrin receptors, in a complex biological matrix. Competitive adsorption experiments coupled with theoretical calculations reveal that the (100) facet of cadmoselite and (002) facet of greenockite preferentially bind with transferrin via inner-sphere thiol complexation. Molecular dynamics simulation infers that facet-dependent transferrin binding is also induced by the differential affinity of crystal facets to water molecules in the first solvation shell, which affects access to exposed facets. Overall, this research underlines the promise of facet engineering to improve the efficacy of crystalline nanomaterials in biological applications.

[1] College of Environmental Science and Engineering, Ministry of Education Key Laboratory of Pollution Processes and Environmental Criteria, Tianjin Key Laboratory of Environmental Remediation and Pollution Control, Nankai University, 38 Tongyan Rd., Tianjin 300350, China. [2] State Key Laboratory of Environmental Chemistry and Ecotoxicology, Research Center for Eco-Environmental Sciences, Chinese Academy of Sciences, Beijing 100085, China. [3] University of Chinese Academy of Sciences, Beijing 100049, China. [4] School of Environment, Beijing Normal University, 19 Xinjiekouwai St., Beijing 100875, China. [5] Department of Civil and Environmental Engineering, Rice University, 6100 Main Street, Houston, TX 77005, USA. ✉email: zhangtong@nankai.edu.cn; alvarez@rice.edu

Crystalline nanomaterials can be conjugated with a variety of biomolecules, including peptides, proteins (e.g., antibodies and enzymes), nucleic acids (e.g., aptamers), and lipids, for enhancing their biocompatibility and in vivo circulation, as well as for enabling cell recognition and intracellular delivery[1–5]. This approach is becoming increasingly prominent for diagnosis and treatment of cancer, anemia, diabetes, and Alzheimer's disease via nano-enabled biosensors, bioimaging, and drug delivery[6–8]. Significant attention has focused on modulating the interaction between crystalline nanomaterials and biomolecules through manipulating the morphology, particle size, and surface functionalities of nanocrystals[9–12], whereas exposed crystal facets, one of the most intrinsic properties of crystalline nanomaterials, remain largely unexplored.

Some biomolecules are known to exhibit differential affinities toward dissimilar crystal surfaces. For example, thermal hysteresis proteins, a group of serum proteins commonly present in organisms living in cold environments, bind to specific faces of ice crystals to enable their antifreeze activity[13,14]. Recent theoretical studies point to the possibility of facet-dependent selective binding of amino acids, peptides, proteins, and DNA to crystal surfaces containing metals[15–19]. Here, we experimentally prove the concept that facet engineering may be utilized to tune the nanocrystal–biomolecule association for refining biological applications of crystalline nanomaterials. Using transferrin-facilitated cellular targeting as a model system (which has been widely applied in cancer-related research[20–22]), we demonstrate that cadmium chalcogenide nanocrystals with specific facets (i.e., (100) facet of cadmoselite and (002) facet of greenockite) preferentially bind with transferrin via inner-sphere coordination in a complex protein matrix, which significantly enhances receptor-mediated delivery of the nanocrystals into cancer cells.

## Results

**Characterization of different-faceted nanocrystals.** Three types of facet-engineered cadmium chalcogenide nanocrystals were used in this study, including cadmium selenide (CdSe) nanoparticles (CdSe-p), CdSe nanorods (CdSe-r), and cadmium sulfide (CdS) nanorods (CdS-r). The crystalline phase of CdSe and CdS was cadmoselite and greenockite, respectively. For each type of nanocrystals, two materials (denoted as "A" and "B") with different content of exposed facets were synthesized to exhibit similar size and morphology. The relative height of the (100) or (101) vs. (002) peaks in the X-ray diffraction (XRD) spectra was used to estimate the relative content of these facets in each material[23]. CdSe-p-A and CdSe-r-A had larger relative content of (100) compared with the corresponding "B" materials, while CdS-r-A had larger relative content of (002) compared with CdS-r-B (Fig. 1a–c). CdSe nanoparticles appeared to be spherical particles with diameter of ~30 nm (Fig. 1d), while the dimensions of CdSe and CdS nanorods were 400 × 20 nm (Fig. 1e) and 100 × 10 nm (Fig. 1f), respectively. The hydrodynamic diameter, $\xi$ potential and Brunauer–Emmett–Teller (BET) surface area of the "A" materials were similar to those of the respective "B" materials (Supplementary Table 1). Thus, the difference in exposed crystal facets was the main factor determining differences in nanocrystal–transferrin binding efficacy and uptake by cancer cells.

**Facet-dependent transferrin binding to nanocrystals.** The facet-engineered cadmium chalcogenide nanocrystals were incubated in a model protein matrix (i.e., fetal bovine serum, FBS) that contained transferrin along with a diverse group of proteins. The composition of the hard protein corona on CdSe-p, CdSe-r, and CdS-r was analyzed using liquid chromatography–mass spectrometry/mass spectrometry (LC–MS/MS, Supplementary Data). The exponentially modified protein abundance index (emPAI) ratio of the protein fractions associated with the nanocrystals with respect to the fractions in FBS (i.e., enrichment factor[24]) was calculated to represent the degree of enrichment of specific proteins in the hard corona. Interestingly, transferrin was the most enriched protein in the corona on all the tested nanocrystals, particularly for the "A" materials, as indicated by the larger enrichment factor of transferrin compared with the other serum proteins (Fig. 2; Supplementary Data). Moreover, among all the proteins in the matrix, transferrin exhibited the greatest difference in the enrichment factor of the "A" vs. "B" samples (Fig. 2; Supplementary Data). These results indicate that Cd nanocrystals bind transferrin preferentially in the presence of other biomolecules, and this binding process is susceptible to facet modulation, which may be exploited to enable specific binding of transferrin for enhanced cell targeting of nanocrystals in a complex and realistic biological matrix.

**Facet-dependent delivery of nanocrystals into cancer cells.** Indeed, both single-cell–inductively coupled plasma–mass spectrometry (SC–ICP–MS, Fig. 3a–d) and confocal fluorescence microscopy analysis (Fig. 3e) of HeLa cells after exposed to the transferrin–CdSe conjugates (transferrin was FITC-labeled for confocal fluorescence microscopy) revealed that the stronger binding between transferrin and CdSe-p-A, relative to CdSe-p-B, resulted in greater uptake of these protein–nanocrystal conjugates into HeLa cells. This facet-dependent cellular uptake was mediated via transferrin receptors, proven by the endocytosis experiments using HeLa cells with transferrin receptors silenced by small interfering RNA (siRNA). The siRNA-transfected cells assimilated much less nanomaterial with no apparent differences between CdSe-p-A and CdSe-p-B (Supplementary Fig. 1). Note that previous research has shown that transferrin–nanomaterial conjugates may lose the targeting function due to the formation of a corona by other biomolecules (abundant in biological environments) on nanomaterial surfaces that hinder the intended function[11]. Our results clearly demonstrate that transferrin molecules that preferentially associated with CdSe nanocrystals with more abundant (100) facet remained active after adsorption, and were recognized by the receptors on the HeLa cells.

## Discussion

Effective application and optimization of this facet-facilitated cell targeting phenomenon calls for mechanistic understanding of the surficial interaction between transferrin and different exposed facets of the cadmium chalcogenide nanocrystals. Nanoparticles that contain soft metals (e.g., Au, Zn) are known to have strong affinity for and tend to adsorb thiol-containing ligands by forming coordination bonds[25,26]. Given that inner-sphere complexation has been shown to be an important mechanism for facet-dependent adsorption of anion ligands on hematite nanocrystals[27,28], it is reasonable to postulate that the facet-dependent binding between transferrin, a thiol-rich protein, and nanocrystals containing soft metal Cd is likely controlled by the metal–thiol complexation process. Accordingly, we demonstrate that the relatively high thiol content is an important factor controlling the facet-dependent preferential binding of transferrin to cadmium chalcogenide nanocrystals by substituting all cysteine for glycine to synthesize a non-thiol mutation of transferrin (Fig. 4a) and comparing this mutation with the thiol-rich transferrin (Cys% 5.39, Fig. 4b) in adsorption experiments. In all cases (i.e., CdSe-p, CdSe-r, and CdS-r), thiol-rich transferrin preferentially bound with the "A" rather than the "B" materials (Fig. 4c–e), consistent with the trend in proteomic analysis

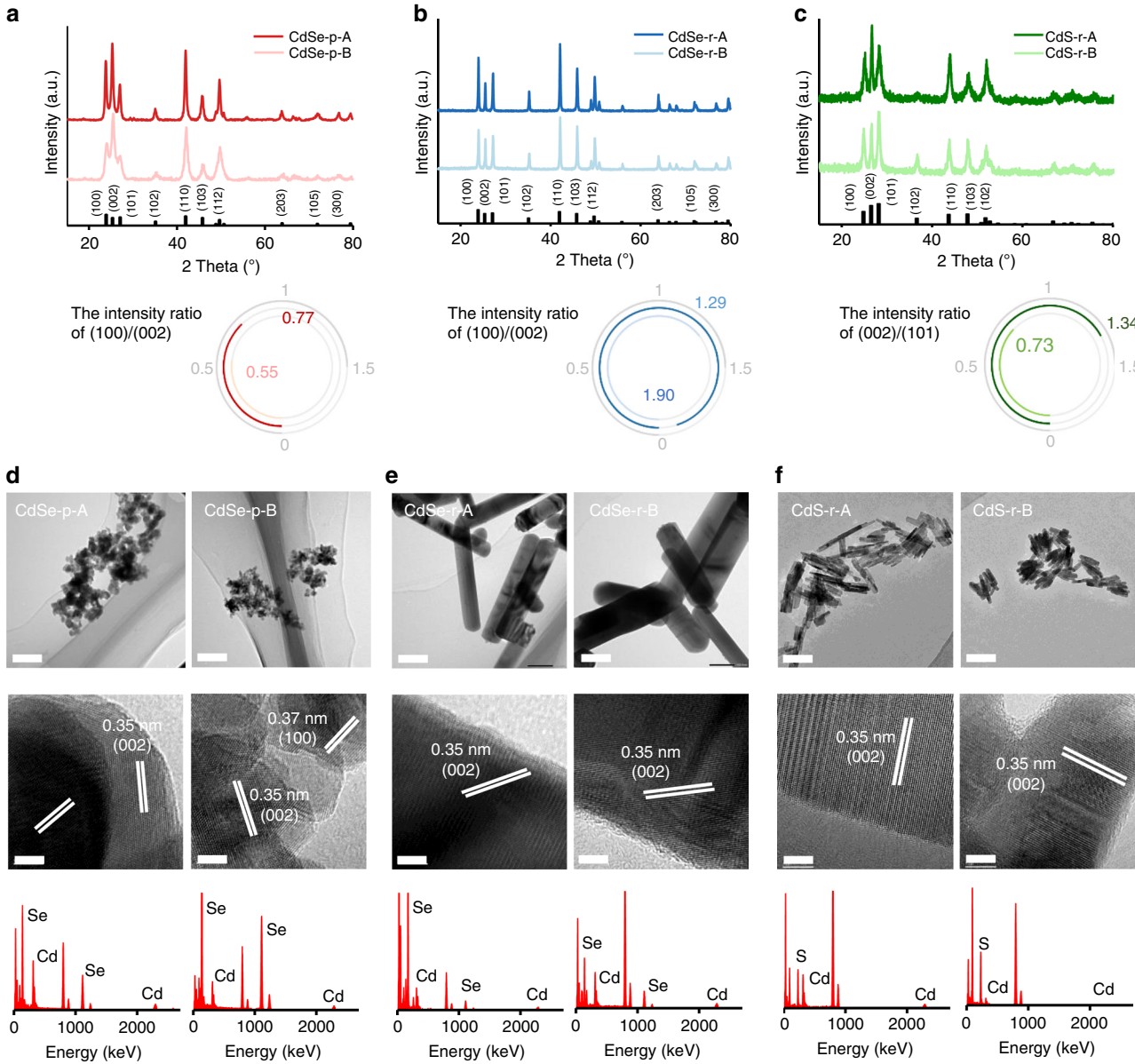

**Fig. 1 Characterization of different-faceted CdSe and CdS nanocrystals. a–c** X-ray diffraction (XRD) spectra of CdSe nanoparticles (**a**, CdSe-p-A: dark red solid line; CdSe-p-B: light red solid line), CdSe nanorods (**b**, CdSe-r-A: dark blue solid line; CdSe-r-B: light blue solid line), and CdS nanorods (**c**, CdS-r-A: dark green solid line; CdS-r-B: light green solid line). The two nanomaterials in each group had different facet content (i.e., (100) vs. (002) for CdSe and (002) vs. (101) for CdS), according to the intensity ratio of the corresponding peaks in the XRD spectra. **d–f** Transmission electron microscopy (TEM) images and energy-dispersive spectroscopy (EDX) spectra showed that different-faceted CdSe nanoparticles (**d**), CdSe nanorods (**e**), and CdS nanorods (**f**) exhibited similar size and morphology. TEM analysis was independently repeated five times, and the results were similar. Scale bar of upper panel: 100 nm. Scale bar of middle panel: 5 nm. Source data are provided as a Source Data file.

(Fig. 2), whereas adsorption of the non-thiol transferrin to the Cd nanocrystals appeared to be independent of or less affected by exposed facets and occurred to a much lower extent than with thiol-rich transferrin (Fig. 4c–e).

To further discern the role of thiols in transferrin binding with different facets, competitive adsorption experiments using a model thiol-rich protein (i.e., bovine serum albumin (BSA)) and low-molecular-weight model compounds were conducted (Fig. 4f), and the experimental data were complemented with theoretical computation (Fig. 4g–l). As expected, BSA similarly exhibited preferential binding toward CdSe-p-A relative to CdSe-p-B (Supplementary Fig. 2). Thiol-containing amino acid, cysteine, and its non-thiol analog, serine, were also compared for their adsorption affinities to CdSe-p-A vs. CdSe-p-B. In the same

reaction matrix, cysteine outcompeted serine and preferentially bound with nano-CdSe, particularly to a greater extent with the "A" material than the "B" material (Fig. 4f). The specific inter-action between thiol and different facets of CdSe were assessed via density functional theory (DFT) calculation, and the favorable adsorption process was indicated by the lower (i.e., more negative) values of the adsorption energy (Fig. 4g–l). All calcu-lated adsorption energies ($-1.62$ to $-0.95$ eV, Fig. 4i, l) were sufficiently negative to indicate chemical adsorption pathways. CdSe-(100) facet exhibited lower adsorption energy toward cysteine compared with CdSe-(002) facet, and CdS-(002) facet exhibited lower adsorption energy toward cysteine compared with CdS-(101) facet. Altogether, these results reveal that trans-ferrin associated with Cd nanocrystals mainly via chemical

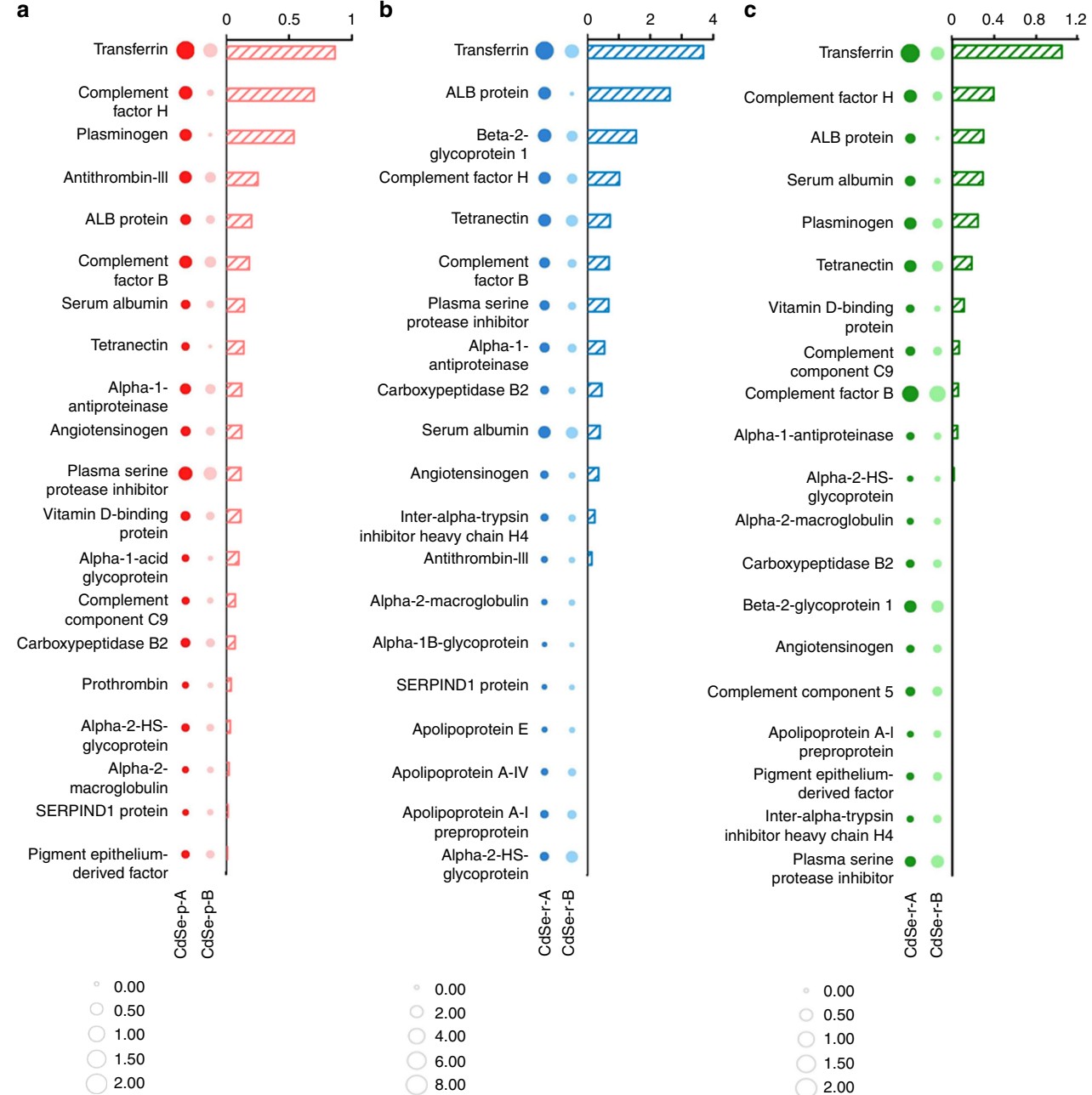

**Fig. 2 Transferrin was enriched in the hard protein corona and the extent of enrichment was susceptible to the exposed facets of CdSe and CdS nanocrystals.** The protein coronas were formed in fetal bovine serum. The size of the circles represented the enrichment factor of proteins in the hard corona, and the difference in the protein enrichment factors between the two different-faceted materials is shown in the bar graph. The proteins were ranked according to the decreasing difference in the enrichment factor of proteins detected from the hard coronas on the two different-faceted CdSe nanoparticles (**a**, CdSe-p-A: dark red bubble; CdSe-p-B: light red bubble), CdSe nanorods (**b**, CdSe-r-A: dark blue bubble; CdSe-r-B: light blue bubble), and CdS nanorods (**c**, CdS-r-A: dark green bubble; CdS-r-B: light green bubble). Source data are provided as a Source Data file.

adsorption, and this process occurred through inner-sphere thiol complexation and depended on the exposed crystal facet.

Molecular dynamics (MD) simulations were conducted to evaluate the interaction between the entire protein molecules with different crystal facets, to complement the DFT calculation that focused on the thiol moieties. In the MD calculation, we established six initial structures of transferrin for 20-ns simulations by rotating the transferrin 90° around the x- and y-axes (Supplementary Figs. 3 and 4), and utilized the most stable structure (i.e., largest contact atom number) for the comparison between (100) and (002) facets of CdSe using 200-ns simulations (Fig. 5a, b). In

order to mimic chemical adsorption process, the transferrin molecules were initially placed 2 nm away from CdSe (100) or (002) surface, and used a spring constant of 1000 kJ mol$^{-1}$ nm$^{-2}$ and a pull rate of 0.1 nm ns$^{-1}$ to accelerate the initial adsorption. A harmonic potential was utilized to restrain the S atoms in position after they directly contacted with the CdSe (100) surface[26]. The number of all protein atoms in direct contact with (100) facet gradually increased from 107 to 430 during the simulation (Fig. 5c), reflecting the thermodynamic feasibility of transferrin adsorption to CdSe-(100). In contrast, transferrin molecules were prone to "fleeing away" from the (002) facet,

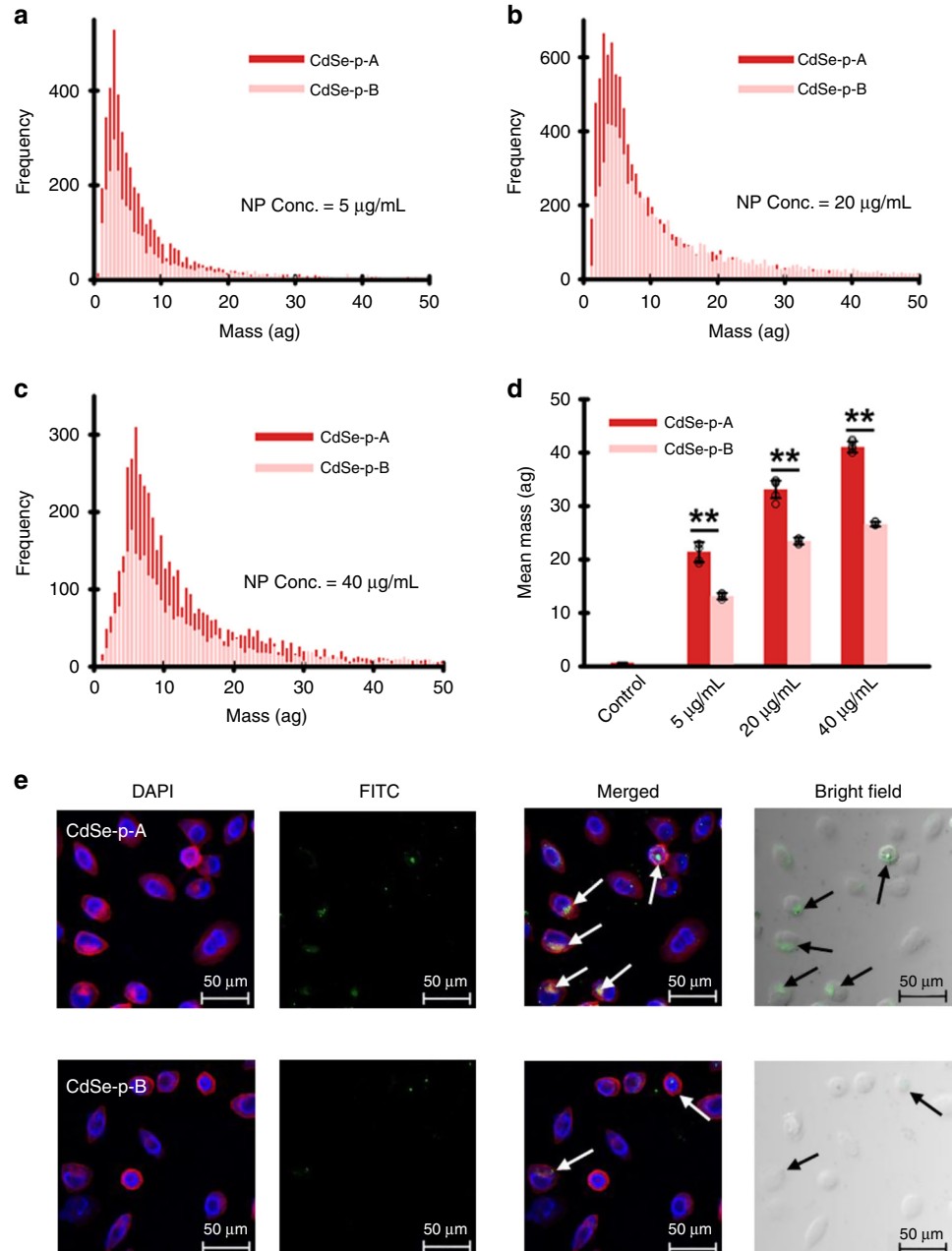

**Fig. 3 Facet-dependent binding of transferrin and cellular uptake of CdSe nanoparticles. a–d** Single-cell–inductively coupled plasma–mass spectrometry (SC–ICP–MS) histogram of mass distribution (**a**–**c**) and mean mass (**d**) of CdSe nanoparticles (CdSe-p-A: dark red bar; CdSe-p-B: light red bar) taken up by HeLa cells, after cell incubation with transferrin–CdSe conjugates for 3 h. Results show higher cellular content of nano-Cd in HeLa cells exposed to "A" materials (i.e., CdSe with more (100) facet) that preferentially bound with transferrin. Data are presented as mean ± SD of five replicate samples ($n = 5$; $p = 0.0007$, $0.0004$, $<0.0001$ for the group of 5, 20, 40 μg/mL CdSe nanoparticles by one-way ANOVA, respectively). **e** Confocal laser scanning microscopy images of HeLa cells after incubated with FITC-labeled transferrin–CdSe conjugates demonstrated greater cellular uptake of "A" materials. Microscopic analysis was independently repeated five times, and the results were similar. Statistical significance between groups: (**) $p < 0.01$. Source data are provided as a Source Data file.

indicated by the slight decrease in the contact atom number (Fig. 5c), and thus the transferrin association with the (002) facet was relatively unstable. The sulfur atoms directly contacted with nano-CdSe were all originated from the disulfides moieties of transferrin molecules, with more sulfur binding on (100), relative to (002) (Fig. 5d). There were six sulfur-binding sites, including C118–194, C171–177, and C158–174, on CdSe-(100) facet, as opposed to only four sulfur-binding sites, including C137–331 and C615–620, on CdSe-(002) facet. The smaller fluctuation in the number of contact sulfur atoms on (100) vs. (002), especially

toward the end of the simulation (Fig. 5d), suggested that thiol binding on (100) was more stable than that on (002).

Recent studies have reported that water layers adjacent to the material surface play an important role in the adsorption of biomolecules[19,29]. Our MD calculations showed that the difference in the number and arrangement of water molecules in the first solvation shell (FSS) on (100) vs. (002) also contributed to the facet-dependent binding affinity of transferrin. The FSS on the (100) facet contained fewer layers and lower density of water molecules (Fig. 5e, g), compared with the FSS on the (002) facet

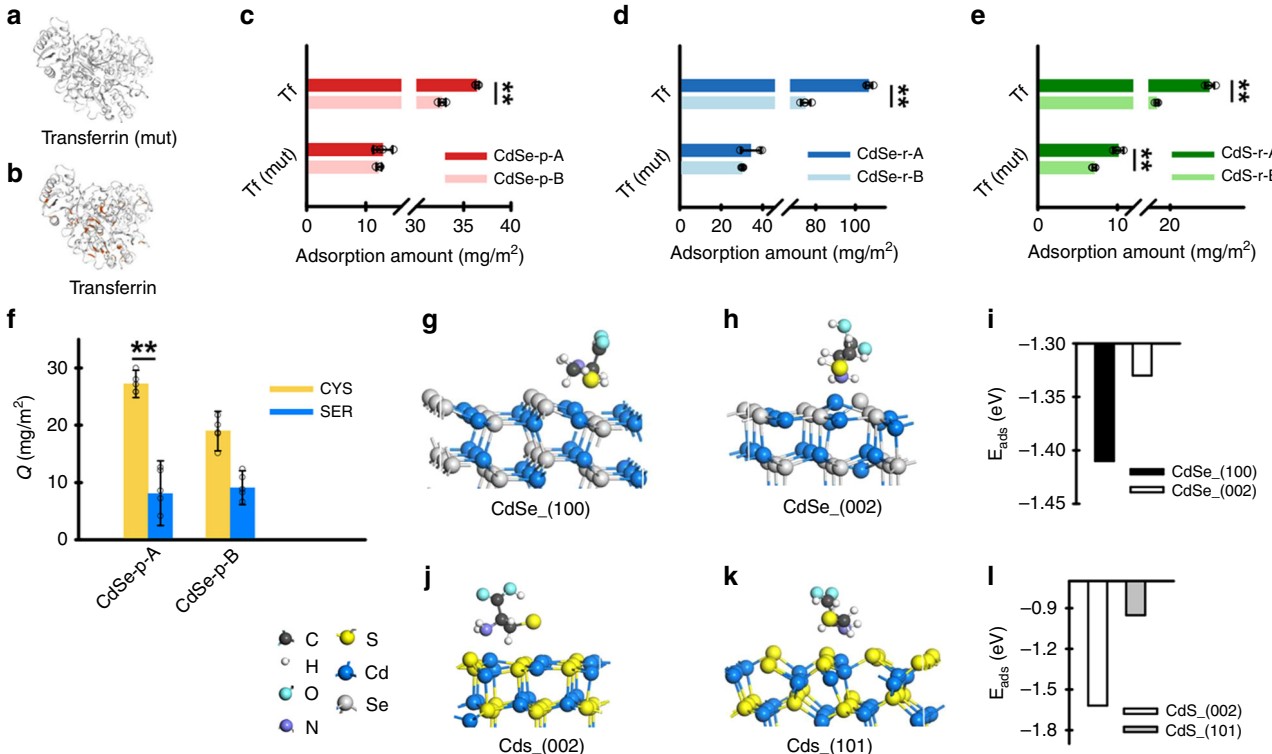

**Fig. 4 Facet-dependent preferential binding of transferrin with Cd nanocrystals occurred through thiol complexation. a, b** Schematic description of the structures of transferrin and non-thiol transferrin (mut) (cysteine shown in red). Adsorption of transferrin on CdSe nanoparticles (**c**, CdSe-p-A: dark red bar; CdSe-p-B: light red bar; $p = 0.0006$ for transferrin and 0.4071 for transferrin (mut)), CdSe nanorods (**d**, CdSe-r-A: dark blue bar; CdSe-r-B: light blue bar; $p < 0.0001$ for transferrin and $=0.1306$ for transferrin (mut)) and CdS nanorods (**e**, CdS-r-A: dark green bar; CdS-r-B: light green bar; $p = 0.0003$ for transferrin and 0.0006 for transferrin (mut)) was consistently greater compared with the non-thiol transferrin (mut). Data are presented as mean ± SD of four replicate samples, and analyzed by one-way ANOVA ($n = 4$). Competitive adsorption of cysteine (CYS) vs. serine (SER) (**f**, CYS: yellow bar; SER: blue bar) on the two different-faceted CdSe nanoparticles showed that thiol-containing compounds preferentially adsorbed to the "A" material. Data are presented as mean ± SD of five replicate samples ($n = 5$; $p = 0.0098$ for cysteine and 0.8567 for serine by one-way ANOVA). Density function theory (DFT) calculation indicated that cysteine exhibited lower adsorption energy toward (100) vs. (002) of CdSe (**g–i**, CdSe_(100): black bar; CdSe_(002): white bar), and exhibited lower adsorption energy toward (002) vs. (101) of CdS (**j–l**, CdS_(002): white bar; CdS_(101): gray bar). Statistical significance between groups: \*\*$p < 0.01$. Source data are provided as a Source Data file.

(Fig. 5f, g). The facet-dependent differential affinity to water was corroborated by the adsorption energy values of water molecules on the CdSe-(100) facet ($-0.33$ eV) vs. CdSe-(002) facet ($-0.66$ eV) according to the DFT calculation. Due to the relatively low affinity of the (100) facet to water, the hydrogen bonds in the FSS on this facet dominantly formed among water molecules (Fig. 5h). In the FSS on the (002) facet, although the overall abundance of hydrogen bonds was smaller than the (100) facet (Fig. 5i, j), a good fraction of hydrogen bonds formed between water molecules and the (002) facet. As a result, the interfacial water was more tightly bound to the (002) facet than to the (100) facet, which rendered it difficult for transferrin to replace these water molecules prior to forming inner-sphere coordination bonds with the (002) facet. Hence, a less compact and loosely bound water molecule network occurred on the (100) facet and facilitated the approaching process of transferrin toward the facet from bulk solution, which may, at least partly, explain the preferential binding of transferrin to CdSe nanocrystals with larger (100) content (Figs. 2 and 4c–e).

Significant research attention has focused on predicting and manipulating the behavior of engineered nanomaterials in biological environment based on their morphology and surface coating. Our research points to a long underappreciated parameter, the exposed crystal facet. We demonstrate the feasibility of facet modulation for enhancing the receptor-mediated cancer cell delivery of cadmium chalcogenide nanocrystals via preferential

and stable chemical adsorption of transferrin to specific exposed facets in a complex biological matrix. This process was controlled by the affinity of the crystal facets for the thiol moieties of proteins as well as for the water molecules in the FSS. Given that chemical complexation is a common mechanism for adsorption of ligand-rich macromolecules onto metal-containing surfaces, the implication of our discovery should not be limited to the case of nano-CdSe and CdS, and may be extended to more biocompatible nanocrystals, particularly those containing soft metals (e.g., Au, Ag, Pt, Pd, and Zn). Such nanocrystal–biomolecule complexes should be examined under realistic scenarios (in vivo) prior to considering biomedical applications.

Overall, many thiol-rich proteins besides transferrin (e.g., serum albumin, complement factor H) mediate important physiological functions in the human body, and likely modulate biological responses to nanomaterials via specific protein–receptor pathways. Therefore, facet engineering of nanocrystals containing soft metals offers a promising approach for future design of metal-containing nanostructures with improved safety and efficiency in biological applications.

## Methods

**Preparation and characterization of nanocrystals**. All chemical reagents used were purchased from Sigma-Aldrich (China). The CdSe nanoparticles were synthesized using the following methods[30]. For synthesizing CdSe nanoparticles, 0.048 g Se powder, 0.133 g Cd(AC)$_2$·2H$_2$O, and 0.2 g NaOH were mixed in 20 mL $n$-butyl

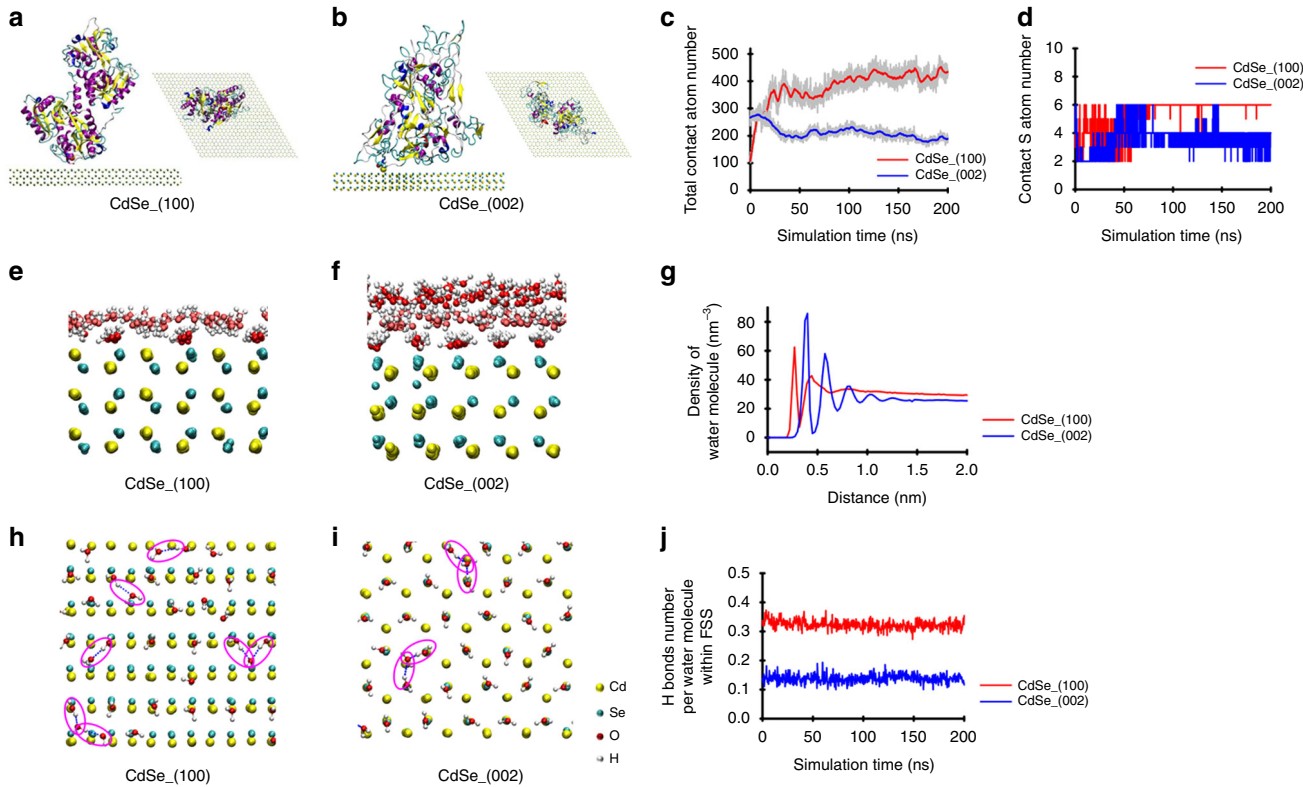

**Fig. 5 Molecular dynamics (MD) simulation of interaction between transferrin with CdSe (100) and (002) facets.** Conformation of transferrin was different on (100) facet (**a**) and (002) facet (**b**), which resulted in different total contact atom number (**c**, CdSe_(100): red solid line; CdSe_(002): blue solid line) and contact sulfur atom number (**d**, CdSe_(100): red solid line; CdSe_(002): blue solid line). Snapshot of water molecule layers (**e**, **f**), number density of water molecules along the z-direction (**g**, CdSe_(100): red solid line; CdSe_(002): blue solid line), and the network (**h**, **i**) and number (**j**, CdSe_(100): red solid line; CdSe_(002): blue solid line) of hydrogen bonds in the first solvation shell of CdSe (100) and (002) facets. The hydrogen bonds were shown as blue dash lines and highlighted in purple circles (**h**, **i**). Source data are provided as a Source Data file.

alcohol, and heated at 150 °C for 11 h. 0.2 g EDTA was added during synthesis of CdSe-p-B nanomaterials. For synthesizing CdSe nanorod A (CdSe-r-A), 3.08 g Cd $(NO_3)_2 \cdot 4H_2O$ was dissolved in 10 ml of DI water, and then $NH_3 \cdot H_2O$ was slowly added into the solution to adjust pH. In another solution, 0.86 g $Na_2SeO_3$ was stirred for 5 min with 15 ml of $N_2H_4 \cdot H_2O$ and then was mixed with the Cd solution. The pH of the solution was adjusted to 11 and heated at 180 °C for 4 h[31]. For CdSe-r-B, 0.039 g Se powder was mixed with hydroxides (NaOH: KOH = 51.5: 48.5), 2 mL $N_2H_4 \cdot H_2O$, and 8 mL DI water. Then 0.154 g $Cd(NO_3)_2 \cdot 4H_2O$ was added, and the solution was heated at 200 °C for 24 h[32]. For synthesizing CdS nanorod A (CdS-r-A), 0.3 g of $CdCl_2$ and 0.4 g of thiourea were mixed with 2 mL of DI water and 18 mL of ethylenediamine. Then the solution was heated at 150 °C for 5 h[23]. For CdS-r-B, 0.266 g $Cd(AC)_2 \cdot 2H_2O$ and 0.079 g L-cysteine were dissolved in 2 ml DI water. Then, 18 mL ethanolamine (EA) was added to the solution, which was then heated at 180 °C for 24 h[33].

X-ray diffraction (XRD) spectra of the synthesized CdSe and CdS nanomaterials were obtained using Rigaku D/Max III diffractometer (D/MAX2500, Rigaku Inc., Japan) with Cu K radiation, $\lambda = 1.5418$ Å. Physical dimensions and morphologies of the materials were characterized by transmission electron microscopy (TEM) coupled with energy-dispersive spectroscopy (EDX) (JEM-2100, JEOL, Japan). The $D_h$ and ξ-potential of the CdSe and CdS nanomaterials were determined using a ZetaSizer (Nano ZS, Malvern Instruments, UK). The Brunauer–Emmett–Teller (BET)-specific surface area was determined by multipoint $N_2$ adsorption–desorption method using an accelerated surface area and porosimetry system (ASAP2010, Micromeritics Co., USA).

**Characterization of protein corona on nanocrystals.** The CdSe and CdS suspension was sonicated (100 W) for 30 min before incubating with 10% (v/v) fetal bovine serum (FBS, Genview, China) at 37 °C. After 4 h, the CdSe/CdS–protein complexes were separated from the supernatant plasma by centrifugation at 12,396 × g (4 °C) for 10 min. The pellet was washed with phosphate buffer saline (PBS) three times to remove the proteins with low affinity for the surface of CdSe/CdS nanomaterials (i.e., the soft protein corona).

The composition of the hard protein corona formed on the surface of nano-CdSe and CdS were characterized using liquid chromatography–mass spectrometry/mass spectrometry (LC–MS/MS). All chemical reagents used here were purchased from Sigma-Aldrich (China). For LC–MS/MS analysis, proteins

were digested in solution containing 20 μL trypsin overnight at 37 °C. The enzymatic hydrolysis was terminated by the addition of 0.1% formic acid. After digestion, the peptide mixtures were analyzed by LC–MS/MS using an ultimate 3000 nanoLC system (Dionex Inc., USA) and a tandem mass spectrometry (MS/MS) in a Q Exactive (Thermo scientific Inc., USA). Peptides were separated using a $C_{18}$ trap column (Thermo scientific Inc., USA) with DI water (A: $H_2O$, 0.1% FA) and acetonitrile (B: ACN, 0.1% FA) as eluents. The MS was operated in the positive ion mode and the m/z ratio of 350–2000 was scanned. For protein identification, raw data files were converted to Mascot generic format (mgf) files and searched in National Center for Biotechnology Information (NCBInr) database via Mascot Search (version 2.3.01, Matrix Science, http://www.matrixscience.com/). Mascot searching parameters included trypsin as the proteolytic enzyme with one missed cleavage. The modifications of methionine oxidation were selected. Peptide charge was set to +1, +2, and +3. A minimal Mascot score of 100 was set for protein identity validation. The 20 most abundant proteins (according to exponentially modified protein abundance index, emPAI values) in hard corona of each material were chosen for comparison between different-faceted materials. The enrichment factor of protein fractions were the emPAI ratio of the protein fractions (normalized with respect to BET surface area) associated with the nanocrystals with respect to the fractions in FBS[24].

**Cell silencing and cellular uptake of nano-CdSe.** HeLa cells were seeded on tissue-culture-treated six-well plates at a density of 300,000 cells per well and incubated at 37 °C (5% $CO_2$) for 24 h to ensure adhesion onto the tissue-culture plate surface. Briefly, 100 pmol small interfering RNA (siRNA) oligo (GenePharma Biotechnology, China) for each well were transfected into cells using Lipofectamine 2000 (Invitrogen) according to the manufacturer's instructions. Two siRNA sequences (siRNA-2423 and siRNA-2061, respectively) were synthesized as follows: for siRNA-2423, sense strand was 5′-GAACUUGAAACUGCGUAAATT-3′ and antisense strand was 5′-UUUACGCAGUUUCAAGUUCTT-3′; for siRNA-2061, sense strand was 5′-GCUGGUCAGUUCGUGAUUAUT-3′ and antisense strand was 5′-UAAUCACGAACUGACCAGCTT-3′. The silencing effects were confirmed by western blot experiments (Supplementary Fig. 1). Cells were transfected with siRNAs for 48 h in all experiments before exposure to CdSe nanoparticles. After silencing, cells were washed by DMEM medium and then replaced by nanoparticle dispersions (i.e., diluted by DMEM with 10% FBS to concentration of

40 µg mL$^{-1}$) and incubated for 3 h before harvesting. The samples were digested by dry baths/block heaters (Thermo Scientific, China), and the cellular content of Cd was quantified using an inductively coupled plasma–mass spectrometer (ICP-MS, Agilent 8800, Agilent Technologies, Inc., China).

**Cellular uptake of nano-CdSe analyzed by SC−ICP−MS.** HeLa cells were cultured in Dulbecco's Modified Eagle's medium (DMEM, Gibco BRL Life Technologies Inc., USA) supplemented with 10% FBS (Gibco BRL Life Technologies Inc., USA) and 100 units mL$^{-1}$ penicillin/streptomycin (Invitrogen, USA) in a humidified 5% CO$_2$-balanced air incubator at 37 °C. For the CdSe nanoparticle uptake experiment, HeLa cells were seeded on tissue-culture-treated 12-well plates at a density of 50,000 cells per well and incubated at 37 °C (5% CO$_2$) for 24 h to ensure adhesion onto the tissue-culture plate surface. The cells were then exposed to CdSe nanoparticles at a concentration of 5, 20, and 40 µg mL$^{-1}$ in DMEM (supplemented with 10% FBS) for 3 h. Next, the cells were washed with PBS five times, enzymatically detached from the tissue-culture plate surface, and fixed in PBS to a final concentration of ~100,000 cells mL$^{-1}$.

All analysis were done on a NexION 2000 inductively coupled plasma–mass spectrometer (ICP−MS) with the Single Cell Application Module in Syngistix software (v 4.0) using the conditions in Supplementary Table 2. The components used for SC−ICP−MS analysis included the NexION 2000 Asperon™ spray chamber as well as the specialized Single Cell Micro DX Autosampler. The Asperon spray chamber was designed to maximize cell transport to the NexION, while the Single Cell Micro DX Autosampler agitated the cellular suspensions prior to analysis to ensure that the cells did not settle from the suspension.

**Cellular uptake of nano-CdSe analyzed by confocal microscopy.** CdSe nanoparticles were incubated with FITC-labeled transferrin (Thermo Fisher Scientific, USA) to form CdSe-transferrin conjugates in PBS. Afterward, HeLa cells were exposed to 20 µg mL$^{-1}$ CdSe-transferrin conjugates for 3 h, washed with PBS three times, fixed with 3.7% paraformaldehyde for 10 min, and then incubated with 0.1% Triton X-100 for 5 min. The cell plasma membranes were stained with rhodamine phalloidin (Solarbio Inc., China), and the cell nuclei were stained with 4',6-diamidino-2-phenylindole (DAPI, Beyotime, China). Fluorescence images were captured using a TCS SP8 laser scanning confocal microscope (Leica, Germany).

**Expression and purification of non-thiol transferrin mutant.** For the expression of the transferrin (mut) proteins, key operational parameters, including sequencing information, are include in Supplementary Table 3. The non-thiol transferrin mutant was synthesized by and available from Beijing Protein Innovation, China. Competent cells of *Escherichia coli.* (BL21) kept at −80 °C were slowly thawed on ice and then incubated with plasmids (pET30a) containing modified genetic sequences (i.e., with cysteines replaced by glycines) and tag (HIS) on ice for 30 min, heat shocked at 42 °C for 90 s, and cooled on ice for 2 min. In total, 800 µL kanamycin-free LB medium was added to the cells, and the cell suspension was incubated at 37 °C for 45 min. After centrifugation at 2152 × *g* for 3 min, the pellets were collected and inoculated onto agar plates containing LB medium with Kanamycin. The plates were dried and incubated at 37 °C overnight.

The colonies were transferred into fresh LB medium with kanamycin and kept at 37 °C on a shaker (200 rpm) until OD$_{550}$ reached 0.6–0.8. Then the cultures were incubated with 0.5 mM of isopropyl β-D-thiogalactoside (IPTG) at 37 °C for another 2 h (200 rpm). After incubation, the colonies were extracted by centrifugation for 1 min (12,396 × *g*), and then lysed by lysis buffer (10 mM Tris-HCl, pH 8.0). The samples were then boiled at 100 °C for 5 min with loading buffer and analyzed by SDS-PAGE to confirm protein expression.

Next, the colonies were transferred into fresh LB medium with kanamycin and kept at 37 °C on a shaker (200 rpm) until OD$_{550}$ reached 0.6–0.8. Then the cultures were incubated with 0.5 mM of IPTG at 16 °C overnight. After incubation, the cell cultures were centrifuged at 5510 × *g* for 6 min, and the pellets were collected and resuspended in 10 mM Tris-HCl (pH 8.0) prior to ultrasonic disruption (500 W, 5-s intervals, 180 cycles). Then, 100 µl of processed culture was centrifuged at 12,396 × *g* for 10 min, and the pelleted proteins were resuspended in 50 µL 10 mM Tris-HCl (pH 8.0) prior to SDS-PAGE analysis, which found that the target proteins were in the culture pellet instead of supernatant.

To purify target proteins, a nickel-affinity column (Ni-Sepharose 6 Fast Flow, GE Healthcare, USA) with immobilized nickel ions was applied after equilibrated with a loading buffer containing 10 mM Tris-HCl (pH 8.0), 8 M urea, and 0.5 M NaCl. The target proteins were intercepted by the column and then eluted with imidazole at gradually increasing concentrations of 15, 60, 500 mM in 10 mM Tris-HCl (pH 8.0), and the purified proteins were collected from eluent. The purity of transferrin (mut) was assessed using SDS-PAGE analysis. After elution, the sample was dialyzed with buffer (1% glycine, 0.1% SDS, 5% glycerine, and 10 mM Tris-HCl, pH 8.0) containing gradient concentration (6 M, 4 M and 2 M, each concentration for 3 h) of urea at 4 °C. Then the sample was dialyzed with buffer containing 1% glycine, 0.1% SDS, 5% glycerine, and 10 mM Tris-HCl (pH 8.0) for another 3 h. After dialyzed with 10 mM Tris-HCl (pH 8.0) for 3 h, the sample was centrifuged at 12,396 × *g* for 10 min, and the target proteins were collected in the supernatant.

**Adsorption experiments using model biomolecules.** For adsorption experiments with low-molecular-weight model compounds, a series of 20-ml vials containing 800 µL of 5 mg mL$^{-1}$ CdSe stock suspension were prepared. Then, amino acids stock solution (in PBS) was added to the vials, and the total volume was adjusted to 10 ml with PBS. The CdSe nanoparticle concentration in these vials was 400 µg mL$^{-1}$ and amino acid concentration was 1000 µg mL$^{-1}$, respectively. The vials were shaken at 150 rpm for 4 h to reach adsorption equilibrium. Afterward, the complexes were separated by centrifugation at 12,396 × *g*, 4 °C for 10 min, and the supernatant was withdrawn to analyze the concentrations of amino acid. The concentrations of cysteine and serine were determined according to the following protocols[34]. Sample aliquots were derivatized by 2,2'-dithiobis(5-nitropyridine) (DTNP, Sigma-Aldrich, China), and cysteine concentration was quantified using a microplate reader (Synergy H4, Bio tek, USA). The total concentration of amino acids was determined using the Total Amino Acid Assay Kit (Nanjing Jiancheng Bioengineering Institute, China), following the manufacturer's instructions. The concentration of serine was calculated based on mass balance.

For adsorption of model proteins (i.e., transferrin, transferrin (mut) and BSA) on CdSe/CdS nanomaterials, same procedures were conducted except that the supernatant was withdrawn to analyze the concentrations of proteins using the Bradford Protein Assay Kit (Sangon Biotech, China), following the manufacturer's instructions. For transferrin adsorption, the concentration of transferrin was 50 µg mL$^{-1}$, and the concentration of CdSe nanoparticles was 100 µg mL$^{-1}$. For BSA adsorption, the concentration of BSA was 100 µg mL$^{-1}$, and the concentration of CdSe nanoparticles was 200, 400, and 800 µg mL$^{-1}$, respectively. The blank (i.e., samples without CdSe) showed no adsorption of proteins to the vials, and pH remained constant during the time course of all experiments. The adsorbed mass at each equilibrium concentration was calculated based on a mass balance approach.

**Density functional theory (DFT) computational methods.** All DFT calculations were implemented using the Vienna Ab-initio Simulation Package (VASP). The electron–ion interaction was described by projector-augmented wave method with a plane-wave cutoff of 420 eV. The exchange and correlation potential were described by Perdew–Burke–Ernzerh (PBE) method[35–37]. The van der Waals force was described by DFT-D2 method of Grimme[38]. Spin polarization was considered in all computations. The electronic structure calculations were employed with a Gaussian smearing of 0.1 eV. For adsorption energy to cysteine on CdSe/CdS surface, the Brillouin zone was sampled with $3 \times 3 \times 1$ Monkhorst–Pack k-point grids, and for adsorption energy to H$_2$O, the Brillouin zone was sampled with $5 \times 5 \times 4$, $2 \times 2 \times 1$ and $2 \times 1 \times 1$, Monkhorst–Pack grids, respectively, for CdSe bulk, and CdSe (002), CdSe (100) surfaces calculations[39]. The optimized lattice parameters of wurtzite CdSe were $a = 4.20$, $b = 4.20$, $c = 6.84$. A $(3 \times 2)$ surface unit cell with a four-layer slab for the (100) surface and a $(3 \times 3)$ surface unit cell with a four-layer slab for the (002) surface were used. All slabs were spaced more than 14 Å perpendicular to the slab surface to avoid artificial interaction due to periodicity. During optimization, the atoms of two layers at the bottom were fixed, and the remaining atoms were relaxed to reach stable configurations. Atoms were optimized until the residual forces were below 0.02 eV Å$^{-1}$. The adsorption energies were calculated by the Eq. (1):

$$E_{ads}A = E_{slab+A} - E_{slab} - E_A \qquad (1)$$

**Molecular dynamics (MD) simulations.** The coordinate of transferrin crystal structure and crystal structure of CdSe was obtained from *RCSB Protein Data Bank* (PDB entry: 3V83)[40], *Materials Project*[41], and other references[42], respectively. Six initial structures of transferrin for simulations were obtained by rotating the transferrin 90° around x- and y-axes. Molecular dynamics simulations were performed with GROMACS 5.0.4 package[43–45] with Amber 99SB-ILDN all-atoms force field[46] in the NPT ensemble. The sulfur atoms in cysteine of transferrin were all in disulfide bonds form. In order to accelerate the binding process of transferrin, a pulling force with a constant pulling speed 1 nm ns$^{-1}$ along the z-direction was applied on the sulfur atoms in disulfide bond close to the CdSe surface. To mimic the chemical adsorption of sulfur atoms on CdSe surface, the disulfide bond was constrained to their position on z-direction once the sulfur atoms were in direct contact with CdSe surface (with a distance smaller than 0.35 nm). For each protein-material simulation group, 20 ns MD simulation was applied, and then two represented (most stable) groups were chosen for longer simulation time (i.e., 200 ns). Temperature was maintained at 300 K by applying the Nose-Hoover thermostat coupling[47]. The cutoff switching function for non-bonded van der Waals interaction started at 1.2 nm and reached zero at 1.35 nm. Particle mesh Ewald[48] summation was used to calculate the long-range electrostatic interactions with a cutoff distance of 1.2 nm for the separation of the direct and reciprocal space. The bond lengths were constrained by linear constraint solver algorithm[49], and periodic boundary conditions were applied in all simulations. Protein and materials were dissolved in simple point charge water molecules[50]. The system was neutralized by sodium and chloride ions. Simulations were carried out with a time step of 2 fs, and data were saved every 4 ps. The free VMD software (1.9.2) was used to visualize simulation results[51].

**Statistical analysis.** All data were represented as the mean ± standard deviation (SD). Statistical analysis was performed using independent one-way ANOVA with

Statistical Product and Service Solutions (SPSS, 20.0). *P*-value less than 0.01 ($p < 0.01$) was considered statistically significant.

**Reporting summary**. Further information on research design is available in the Nature Research Reporting Summary linked to this article.

## Data availability

The data that support the findings of this study are available from the authors on reasonable request. The coordinate of transferrin crystal structure was obtained from *RCSB Protein Data Bank* (PDB entry: 3V83, https://www.rcsb.org/structure/3V83). The source data underlying Figs. 1a–c, 2a–c, 3a–d, 4c–f, i–l, 4c, d, g, j and Supplementary Figs. 1–4 and Table 1 are provided as a Source Data file.

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

## Acknowledgements

This project was supported by the National Natural Science Foundation of China (Grants 21976095 and 21425729), Tianjin Municipal Science and Technology Commission (17JCYBJC23100), and the 111 Program of the Ministry of Education of China (T2017002). Partial funding for P.J.J.A. was provided by the NSF ERC for Nanotechnology-Enabled Water Treatment (EEC-1449500). We thank Lu Liu and Xue Chen for assistance with nanomaterial preparation, Haijun Zhang and Weichao Wang

for helping with theoretical calculations, and Ligang Hu, Guangbo Qu, and Qilin Yu for helpful discussion regarding proteomics.

## Author contributions

Y.Q. carried out all experiments and data analysis. T.Z., W.C., and P.J.J.A. conceived the study and supervised the research. C.J., S.L., and C.Z. contributed intellectual input to this study, and Y.Q., T.Z., P.J.A., and W.C. drafted the paper.

## Competing interests

The authors declare no competing interests.
