## [Peer Review File · Nature Communications]

Reviewers' comments:

Reviewer #1 (Remarks to the Author):

Since the submission to Nature Nanotech the authors have done some additional experiments and give a detailed rebuttal to all points provided.

I would like to give the following comments.

1. I see that the authors have done some additional experiments concerning the transferrin hypothesis and appreciate that. The amount of experiments is not overwhelming but I would feel that together with the previous results it is good enough to be published.
2. The authors should cite more work to demonstrate that the phenomenon which is known in principal for e.g. anti-freeze peptides and proteins, i.e. adsorption of peptides or proteins may depend on the facet of the crystal. This would clear the novelty of the paper.
3. The authors did not use serum albumin, complement factor H etc. which I find peculiar as it could have been easily done.
4. Suppl. Table 1: was the zeta potential measure with or without proteins? I would suggest that the authors may also give the zeta potential with proteins if they have this piece of information (should be always around -20 to -30 mV).
5. The substitution of cysteine to serine was not done by the authors which again I find that it could have been done.

In essence I would recommend a minor revision of the paper before final submission to Nature Communications. I would then see that chances are good that the paper can be published.

Reviewer #2 (Remarks to the Author):

The authors have improved the manuscript, but important questions raised by reviewer #2 and reviewer #3 have not been experimentally addressed. In addition, while the authors added some materials' characterization, the different samples are still different, apart from their facets. As the particles are relatively large, an easy experiment would be to use bulk crystals with different facets,

coat them with transferrin, and see if cells attach differently. This would go in line with differences in particle uptake

Reviewer #3 (Remarks to the Author):

The manuscript has been improved and new data have been included. It is appropriate for Nature Communications and should be published with only minor changes.

In the point by point response, most of the comments identified by the reviewers have been addressed convincingly. However, these arguments have not been reflected in the actual revisions of the manuscript. The points made in the rebuttal letter (e.g., limitations on the protein choice, limitations related to Cd NPs, limitations to the in vitro work) are important to provide context and will help the interested reader to better understand the scope of this study. They thus should find their way into the revised manuscript.

Manuscript ID: NCOMMS-19-39044-T

Title: Nanocrystal facet modulation to enhance transferrin binding and delivery into cancer cells

Authors: Yu Qi, Tong Zhang, Chuanyong Jing, Sijin Liu, Chengdong Zhang, Pedro J. J. Alvarez, Wei Chen

Below are our responses to the reviewers' comments. The revised text is shown in red font. To comply with the editorial & publishing policies, we added key operational parameters, including sequencing information, for the expression of the non-thiol transferrin mutant (line 312-314, Supplementary Table 3), and the data availability statement (line 421-425) in the revised manuscript.

Responses to Reviewer 1:

Comment: Since the submission to Nature Nanotech the authors have done some additional experiments and give a detailed rebuttal to all points provided.

I would like to give the following comments.

1. I see that the authors have done some additional experiments concerning the transferrin hypothesis and appreciate that. The amount of experiments is not overwhelming but I would feel that together with the previous results it is good enough to be published.

Response: We thank the reviewer for the positive comments and additional suggestions.

Comment: 2. The authors should cite more work to demonstrate that the phenomenon which is known in principal for e.g. anti-freeze peptides and proteins, i.e. adsorption of peptides or proteins may depend on the facet of the crystal. This would clear the novelty of the paper.

Response: Our research provides the first line of experimental evidence to demonstrate the feasibility of facet modulation for enhancing protein binding and cellular delivery of nanocrystals via protein-receptor pathway in a complex biological matrix. Nevertheless, we agree that some biomolecules may exhibit differential affinities toward different crystal surfaces. Therefore, we added references, particularly regarding the anti-freeze proteins, and revised the text accordingly (lines 49-54):

“Some biomolecules are known to exhibit differential affinities toward dissimilar surfaces. For example, thermal hysteresis proteins, a group of serum proteins commonly present in organisms living in cold environment, bind to specific faces of ice crystals to enable their antifreeze activity.^{13,14} Recent theoretical studies point to the possibility of facet-dependent selective binding of amino acids, peptides, proteins and DNA to crystal surfaces containing metals.^{15-19,}”

Comment: 3. The authors did not use serum albumin, complement factor H etc. which I find peculiar as it could have been easily done.

Response: We used transferrin as a model biomolecule to discern the mechanism of the “facet-dependent” phenomenon during nanocrystal-biomolecule interactions for three reasons. First, transferrin has extensive biological applications, particularly in cancer-related research. Second, transferrin was the most enriched protein in the corona on all the tested nanocrystals in the complex serum matrix (Figure 2). Third, transferrin binding is known to significantly affect the cellular uptake of and responses to nanocrystals via receptor-mediated pathways. Therefore, using transferrin as a model protein helps substantiate an important implication of our study, which is that facet-engineering offers an opportunity to modulate cellular processes by tuning the nanocrystal-protein interactions and the subsequent protein-receptor interactions.

We agree that examining our center hypothesis of facet-dependent nanocrystal-biomolecule binding with another thiol-rich protein will strengthen our paper, and thank the reviewer for this suggestion. Hence, we conducted additional experiments of serum albumin adsorption to different-faceted nanocrystals, which showed consistent facet-dependent trend as in the adsorption experiments using thiol-rich transferrin. These data were

added as Supplementary Figure 2. The corresponding method details were added to the method section (line 362-368), and the data discussion was added to line 132-136 in the main text.

*“To further discern the role of thiols in transferrin binding with different facets, competitive adsorption experiment using a **model thiol-rich protein** (i.e., **bovine serum albumin (BSA)**) and low-molecular-weight model compounds were conducted (Fig. 4(f)), and the experimental data were complemented with theoretical computation (Fig. 4(g-l)). **As expected, BSA similarly exhibited preferential binding toward CdSe-p-A relative to CdSe-p-B (Supplementary Figure 2).**”*

Comment: 4. Suppl. Table 1: was the zeta potential measure with or without proteins? I would suggest that the authors may also give the zeta potential with proteins if they have this piece of information (should be always around -20 to -30 mV).

Response: We added the zeta potential measurements of all the nanocrystals with proteins to Supplementary Table 1; the average values range from -20.3 to -25.1 mV.

Comment: 5. The substitution of cysteine to serine was not done by the authors which again I find that it could have been done.

Response: We agree that serine is an appropriate ‘non-thiol’ analog of cysteine. Thus, we compared cysteine versus serine in a competitive adsorption experiment (Figure 4f, line 137-140) to demonstrate the importance of thiol functional groups in the facet-dependent preferential binding process.

“Thiol-containing amino acid, cysteine, and its non-thiol analog, serine, were also compared for their adsorption affinities to CdSe-p-A vs. CdSe-p-B. In the same reaction matrix, cysteine outcompeted serine and preferentially bound with nano-CdSe, particularly to a greater extent with the “A” material than the “B” material (Fig. 4(f)).”

For the protein adsorption experiments (Figure 4a-e), we replaced cysteine

with another non-thiol amino acid (i.e., glycine), mainly due to the high torsional flexibility of glycine. This experimental design helps maintain the protein configuration, so that it does not overshadow the effect of thiol groups on protein binding.

Responses to Reviewer 2:

Comment: The authors have improved the manuscript, but important questions raised by reviewer #2 and reviewer #3 have not been experimentally addressed.

Response: We thank the reviewer for recognizing our improvements, and note that we conducted additional experiments using transferrin-receptor-silenced cancer cells, which showed significantly lower cellular uptake of the nanocrystals with no facet-dependent variation (Supplementary Figure 1). These additional data provide direct evidence linking the facet-dependent biomolecule binding to the enhanced cellular uptake of the nanocrystals, and address the main concern of the reviewers.

Comment: In addition, while the authors added some materials' characterization, the different samples are still different, apart from their facets.

Response: We include three types of nanocrystals that are different in crystalline phase (i.e., cadmoselite and greenockite) and shape (i.e., nanoparticle and nanorod), to demonstrate that the facet-dependent phenomenon is not limited to one specific type of nanocrystal. However, these three different types of nanocrystals were not compared with each other in our study. Rather, for each of the three types of nanocrystals, two model materials with different exposed facets were synthesized and directly compared. These two model materials were similar in size, shape (Figure 1 d-f), surface charge and state of aggregation (Supplementary Table 1). Therefore, these parameters are not expected to cause the observed differential protein binding or cellular uptake of the two different-faceted materials. This information is included in line 63-68 and line 74-78.

“Three types of facet-engineered cadmium chalcogenide nanocrystals were used in this study, including cadmium selenide (CdSe) nanoparticles (CdSe-p), CdSe nanorods (CdSe-r) and cadmium sulfide (CdS) nanorods (CdS-r). The crystalline phase of CdSe and CdS was cadmoselite and greenockite, respectively. For each type of nanocrystals, two materials (denoted as “A” and “B”) with different content of exposed facets were synthesized to exhibit similar size and morphology.”

“The hydrodynamic diameter, ζ potential and Brunauer–Emmett–Teller (BET) surface area of the “A” materials were similar to those of the respective “B” materials (Supplementary Table 1). Thus, the difference in exposed crystal facets was the main factor determining differences in nanocrystal–transferrin binding efficacy and uptake by cancer cells.”

Comment: As the particles are relatively large, an easy experiment would be to use bulk crystals with different facets, coat them with transferrin, and see if cells attach differently. This would go in line with differences in particle uptake

Response: We kept the model materials within the size range that allowed direct cellular uptake, which is important to demonstrate that facet-dependent preferential binding of biomolecules can be utilized to modulate the cellular responses (not only cell attachment) to nanocrystals. But we agree that the experiment proposed by the reviewer would help examine the binding between cells and the nanocrystal-transferrin complexes. We believe this binding process is essential for cellular uptake, and thus interrogated this binding process in the experiments using transferrin-receptor-silenced cancer cells (Supplementary Figure 1) to explicitly show that the facet-dependent cellular uptake of nanocrystal-transferrin complexes occurred through binding with the transferrin receptors on the cell surface.

Response to Reviewer #3:

Comment: The manuscript has been improved and new data have been included. It is

appropriate for Nature Communications and should be published with only minor changes.

In the point by point response, most of the comments identified by the reviewers have been addressed convincingly. However, these arguments have not been reflected in the actual revisions of the manuscript. The points made in the rebuttal letter (e.g., limitations on the protein choice, limitations related to Cd NPs, limitations to the in vitro work) are important to provide context and will help the interested reader to better understand the scope of this study. They thus should find their way into the revised manuscript.

Response: We appreciate the reviewer's positive comments, and agree that our study has limitations (as well as implications for further research) that should be explicitly recognized. This is now reflected in the revised paper (lines 197-208).

“Given that chemical complexation is a common mechanism for adsorption of ligand-rich macromolecules onto metal-containing surfaces, the implication of our discovery should not be limited to the case of nano-CdSe and CdS and may be extended to more biocompatible nanocrystals, particularly those containing soft metals (e.g., Au, Ag, Pt, Pd and Zn). Such nanocrystal-biomolecule complexes should be examined under realistic scenarios (in vivo) prior to considering biomedical applications.”

“Overall, many thiol-rich proteins besides transferrin (e.g., serum albumin, complement factor H) mediate important physiological functions in the human body, and likely modulate biological responses of nanomaterials via specific protein–receptor pathways. Therefore, facet engineering of nanocrystals containing soft-metals offers a promising approach for future design of metal-containing nanostructures with improved safety and efficiency in biological applications.”

REVIEWERS' COMMENTS:

Reviewer#1: (Remarks to the Author)

The authors have addressed all my concerns/suggestions in full. I recommend to publish this paper as is.

Reviewer#2: (Remarks to the Author)

The revised manuscript should be now accepted.

Responses: We appreciate the reviewer's recommendation to publish this paper.